# An Awareness, Courage, and Love Online Group Intervention for Chinese Older Adults in the Post-Pandemic Era: Study Protocol for a Randomised Controlled Trial

**DOI:** 10.3390/healthcare12212158

**Published:** 2024-10-30

**Authors:** Stephen Cheong-Yu Chan, Qi-Lu Huang, Wing-Shan Ho, Rachel Chan, Crystal Yeung, Serena Wong, Mavis Tsai

**Affiliations:** 1Felizberta Lo Padilla Tong School of Social Sciences, Saint Francis University, Hong Kong, China; wwsho@sfu.edu.hk; 2Department of Social and Behavioural Sciences, City University of Hong Kong, Hong Kong, China; qlhuang5-c@my.cityu.edu.hk; 3Private Practice, Markham, ON L6G 0E8, Canada; h9521793@connect.hku.hk; 4Department of Applied Social Sciences, The Hong Kong Polytechnic University, Hong Kong, China; 5Department of Psychiatry, University of Western Ontario, London, ON N6C 0A7, Canada; swong735@uwo.ca; 6Department of Psychology, University of Washington, Seattle, WA 98195, USA; mavis@uw.edu

**Keywords:** awareness courage and love model, post-pandemic, older adults, social connection, psychological interventions

## Abstract

*Background.* Social connections not only contribute to psychological and emotional well-being but also positively impact physical health, with social isolation and loneliness linked to early mortality and detrimental health outcomes. *Objectives.* This study aims to evaluate an online Awareness, Courage, and Love (ACL) group intervention designed to enhance social connectedness and subjective well-being while reducing loneliness, depression, and anxiety among older adults. *Methods.* This randomised controlled trial (RCT) will involve an intervention group receiving a 2 h ACL session via videoconferencing. Eligible participants selected after screening will be assigned randomly to either a treatment or waitlist-control group. Participants in both conditions will complete the assessments at three time points, including baseline, post-intervention, and one-month follow-up. Tools for assessing the concerned variables include the Inclusion of Other in the Self Scale, de Jong Gierveld Loneliness Scale, Patient Health Questionnaire-9 (PHQ-9), Generalised Anxiety Disorder 2-item (GAD-2) Scale, and World Health Organisation—Five Well-Being Index (WHO-5). *Conclusions.* This study will offer a robust framework for investigating the impact of the online ACL intervention on loneliness and social connectedness among Chinese older adults. The result of this study will reach theoretical, empirical, practical, and teaching significance on mental health care strategies for Chinese older adults.

## 1. Introduction

In recent decades, the global burden of mental disorders has surged. Existing mental health care models strongly focus on diagnosing and treating psychiatric illnesses by highly trained specialists, leaving a void in promotive and preventive strategies that tap into diverse community resources [1,2].

The COVID-19 pandemic has exacerbated mental health challenges for populations of all ages, especially for older adults. Older adults at higher risk of severe illness and death from the virus were found to have a significant fear of COVID-19 [3]. A cross-sectional study of Chinese older adults showed that after the pandemic control measures were relaxed, older adults’ levels of loneliness, anxiety, and depression were higher than those who were surveyed before the relaxation of restriction policies [4]. Older adults might need to make extra efforts to adjust their lifestyle and adapt to the environment after the policy change. These findings suggest an urgent need for expanded mental health care interventions tailored to the unique needs of older adults in the post-pandemic era.

Human capacity for prosocial and cooperative behaviour is unique to a degree among animals. Children typically begin to behave in prosocial ways before two years of age, including comforting others in distress and helping adults by bringing or pointing to out-of-reach objects [5]. Complex prosocial human behaviours, such as verbal language, music, and sports, are believed to have substantially contributed to our evolutionary advantage [6]. Social connections not only contribute to psychological and emotional well-being but also positively impact physical health, with social isolation and loneliness linked to early mortality and detrimental health outcomes [7,8,9,10,11].

The literature on human development and ageing provides several lifespan theories that help describe the changes and resulting outcomes of the ageing process. Socioemotional Selectivity Theory (SST) is one of the lifespan theories that helps explain the shift in personal goals and behaviours with age [12]. This theory posits that older adults (i.e., around age 60) become more focused on goals concerning emotional regulation and emotional meaning (e.g., enhancing psychological well-being). Thus, as individuals grow older, they are particularly selective with respect to whom they choose to keep in their social network, and they tend to cultivate more intimate relationships.

According to SST, older adults are less interested in new social contacts because information obtained by social interactions is less likely to be novel and, thus, less likely to be as valuable [13,14,15,16]. If older adults are not satisfied with the network, they might develop emotional problems. Recent studies have provided evidence to disconfirm that by exploring the role of the use of technology in older adults. For example, Chiarelli and Batistoni [17] investigated the extent to which the postulates of SST can be applied to the social relations of older adults mediated by the use of Facebook. The results indicated that measures of SST were not associated with well-being, which suggests that older adults may not necessarily downsize their social networks online. The potential use of technology to enhance social connectedness in older adults is feasible.

Functional Analytic Psychotherapy (FAP) is a therapeutic approach based on empirically supported behavioural principles that harness the power of the client–therapist relationship to help individuals enhance their social connections and to have more meaningful lives [18]. Evidence for the impact of FAP on various social functioning targets has accumulated across multiple case reports, single-subject designs, and group trials [19]. Therapeutic awareness, courage, and love are essential repertoires in FAP therapists as well as target behaviours in their clients [20]. Derived from FAP [19], the Awareness, Courage, and Love (ACL) Global Project is a non-profit initiative launched by the co-founders of FAP to bring FAP principles to a broader population. In the ACL model, awareness is defined as engaging in mindful awareness of one’s self (feelings, needs, values), other individuals, and the context in which interactions are taking place. Courage is defined as engaging in authentic, vulnerable self-disclosures (e.g., struggles, appreciation), and asking for what one wants and needs. Love is defined as providing empathically accurate responsiveness, including the provision of safety, validation, and giving the other person what they have asked for when possible [21]. The ACL Global Project has grown since its inception in 2015 into over 90 chapters with leaders from 26 countries to facilitate social connections and a sense of belonging for over 10,000 participants [22]. The project’s commitment to evidence-based practice is underscored by ongoing research encompassing laboratory evaluations and randomised controlled trials, aimed at assessing the effectiveness of ACL interventions. Such empirical endeavours serve as the backbone of the project, ensuring that its strategies are informed by scientific evidence [21,23,24,25].

As reviewed, the SST proposes that older individuals may invest more time or resources in nurturing significant relationships. The current approach of ACL interventions provides older adults with a platform to reflect on their relationships and cultivate emotional connections with peers, potentially reducing feelings of loneliness. The ‘Awareness’ component of the ACL model is in line with the enhanced emotional regulation skills emphasised in SST for older adults. The ‘Courage’ aspect of ACL is associated with the self-disclosure required to form deeper, more meaningful relationships within the constrained social networks of older individuals, as outlined in SST. The ‘Love’ element of ACL directly links to the pursuit of emotional fulfilment and overall well-being in older adults, as suggested by SST.

Studies have shown that a well-planned single-session intervention can be as significant as a long-term one [26]. For example, Peters et al. [27] ran a single-session intervention on students and found that a brief positive-future-thinking exercise, where participants wrote about their best possible self and engaged in mental imagery, significantly increased positive affect and future expectancies compared to a control group [27]. Moreover, a recent systematic review supported that single-session interventions may be effective in enhancing mental health in adults, specifically depressive symptoms [28].

Regarding the psychological interventions for well-being in healthy older adults, researchers conducted a systematic review and meta-analysis, and the results supported that those interventions are effective, with a large effect size observed [29]. While a recent study [21] has established the effectiveness of a brief online ACL intervention in helping couples enhance closeness, given that there has been a call for action to help older adults in the post-pandemic era, taken together, we propose to fill the research gap to explore whether an online-version ACL can produce a positive effect on the psychological well-being of the older adult population.

## 2. Study Aims and Hypotheses

This proposed study examines the feasibility, acceptability, and preliminary efficacy of a psychological intervention rooted in the Awareness, Courage, and Love (ACL) model for community-dwelling older adults in Hong Kong. The objectives of this study are as follows:Examine the practical implementation of delivering the ACL intervention to community-dwelling Chinese older adults in Hong Kong, focusing on logistical considerations and potential barriers.Assess the reliability and efficacy of data collection procedures in capturing relevant outcomes and measures within the context of the pilot study.Investigate the preliminary effectiveness of a one-session online ACL intervention in addressing loneliness and depression/anxiety symptoms and promoting social connectedness and subjective well-being among older adults.Determine the magnitude of the effects observed from the ACL intervention on loneliness, anxiety, depression, and improvements in social connectedness and subjective well-being in community-dwelling Chinese older adults.Engaging in new social contacts within the supportive context of an ACL session may hold significant emotional meaning, aligning with older adults’ goals of enhancing psychological well-being.

We hypothesise the following:6.Following participation in the ACL intervention, participants in the treatment condition will report significantly higher levels of (a) social connectedness and (b) subjective well-being, as well as decreased levels of (c) depressive symptoms and (d) loneliness as compared to pre-treatment baseline values and a control group.7.Improved post-intervention levels of (a) social connectedness, (b) subjective well-being, (c) depressive symptoms, and (d) loneliness will remain stable one month following the ACL intervention.

## 3. Materials and Methods

### 3.1. Trial Design

This prospective, two-armed, parallel-group, assessor-blinded, randomised controlled trial (RCT) will assess the effectiveness of an online ACL intervention in reducing loneliness and enhancing social connectedness among community-dwelling Chinese older adults in comparison to a waitlist control group. Data will be collected at three time points: *t*_0_ (baseline, before the intervention), *t*_1_ (post-intervention), and *t*_2_ (one-month post-intervention follow-up). Participants will undergo a comprehensive baseline assessment before the ACL intervention session (*t*_0_). Immediately following the ACL session, participants will complete post-intervention measurements (*t*_1_), mirroring the baseline assessments. One month after the intervention, participants will undergo follow-up measurements (*t*_2_), mirroring the baseline assessments. This follow-up aims to capture the sustainability of the intervention effects. All assessments will be conducted through online surveys, maintaining participant confidentiality.

### 3.2. Participants

This study aims to recruit approximately seventy community-dwelling Chinese older adults through referrals from community organisations. Since the sampling frame of the technological use in Chinese older adults is uncertain, purposeful convenience sampling and snowball sampling will be used for better participant recruitment.

The inclusion criteria for participants will be the following: (a) aged 60 years or older and dwelling in the community; (b) able to communicate in Cantonese; (c) have access to the internet and a videoconferencing device such as a mobile phone; and (d) have the capacity to provide informed consent. The exclusion criteria will be the following: (a) severe emotional distress or presenting an imminent suicidal risk; (b) known history of autism, intellectual disability, schizophrenia-spectrum disorder, bipolar disorder, Parkinson’s disease, or dementia/significant cognitive impairment; (c) illiterate; and (d) difficulty in communication such as aphasia. Older adults who are excluded due to severe emotional distress or active suicidality will be referred to the Integrated Community Centre for Mental Wellness (ICCMW) for treatment. Additionally, emergency contacts and follow-up will be provided by our Co-I and facilitator, who are professional practitioners in counselling. Participants who complete the surveys at all three time points will receive monetary incentives as a token of appreciation. Potential participants will be screened by a research assistant, who will obtain informed consent from eligible older adults.

### 3.3. Interventions

The ACL intervention will consist of a single 120 min online session delivered in a group format via a videoconferencing application (e.g., Zoom). The theme of the session will evolve around “making friends and being seen” to reinforce the appetitive functions of social connection. The intervention will be delivered in a single session. One trial session utilising this clinical protocol was conducted in January 2024 with participants of the Hong Kong Chapter of the ACL Global Project. The clinical protocol is described in Table 1.

### 3.4. Outcomes

Participants will be assessed for the primary and secondary outcomes, in addition to demographic information, including age, gender, marital status, educational level, employment status, and self-rated health status.

#### 3.4.1. Primary Outcome

##### Inclusion of Other in the Self (IOS) Scale

The IOS scale is used to measure how connected a person feels with another individual or group. The original scale includes seven pairs of congruent circles labelled “Self” and “Other”, with varying degrees of overlap ranging from two separate circles to almost completely overlapping circles. Participants select the pair of circles that best represents the closeness of their interpersonal relationship [30]. Scores range from 1 (no overlap) to 7 (most overlap). This study will use an adaptation of the scale in which the circle images are replaced with human figures to make it easier for older adults to comprehend and visualise the measure of closeness (see Appendix A).

#### 3.4.2. Secondary Outcomes

##### 6-Item de Jong Gierveld Loneliness Scale

The de Jong Gierveld Loneliness Scale is a reliable and valid tool used to measure overall, emotional, and social loneliness in older adults [31]. It consists of six items—three negatively formulated items to assess emotional loneliness (“*I experience a general sense of emptiness*”, “*I miss having people around*”, and “*Often, I feel rejected*”) and three positively formulated items to measure social loneliness (“*There are plenty of people that I can lean on in case of trouble*”, “*There are many people that I can count on completely*”, and “*There are enough people that I feel close to*”). Respondents choose from three response categories: “no”, “more or less”, and “yes”. For the negatively worded items, neutral and positive responses are scored as “1”, while for the positively worded items, neutral and negative responses are scored as “1”. This yields a possible score range of 0 (least lonely) to 6 (most lonely). The Chinese version of the 6-item de Jong Gierveld Loneliness Scale has been demonstrated to be a reliable and valid measure of loneliness in Chinese elders, with a Cronbach’s alpha of 0.76 [32].

##### Patient Health Questionnaire (PHQ-9)

The PHQ-9 is a self-report tool commonly used in primary care and non-clinical settings to screen for depression and measure the severity of depression over the last two weeks [33]. It was developed to measure the criteria for major depression based on the DSM-IV. The tool consists of nine items related to anhedonia, sleep issues, depressed mood, fatigue, guilt/worthlessness, appetite changes, difficulty concentrating, suicidal thoughts, and slowed down/restlessness [33]. Respondents rate the items on a 0–3 scale, with “0” indicating “not at all” and “3” indicating “nearly every day”. Total scores for depression can range from 0 to 27, with scores of 5, 10, 15, and 20 representing mild, moderate, moderately severe, and severe depression, respectively [33]. The Chinese version of the PHQ-9 has been shown to be a valid and efficient tool for screening depression. The internal consistency reliability of the Chinese version of the PHQ-9 was 0.86 for the entire scale, indicating a high level of reliability in measuring depression symptoms [34].

##### Generalised Anxiety Disorder 2-Item (GAD-2) Scale

The GAD-2 scale is a brief screening tool for generalised anxiety over the past two weeks. It is a shortened version of the GAD-7 scale, using only the first two questions which represent core anxiety symptoms. The GAD-2 scale has been validated in several studies and has retained the excellent psychometric properties of the GAD-7 scale. It was proposed as a necessary first step for screening generalised anxiety disorder due to its discriminant capability [35]. Possible scores range from 0 to 6, with higher scores indicating higher levels of anxiety symptoms. A score of 3 or more is indicative of possible anxiety. The GAD-2 scale has been found to have acceptable properties for identifying Generalised Anxiety Disorder (GAD) at a cutoff of 3 in the Chinese rural population. The GAD-2 scale demonstrated a Cronbach’s alpha of 0.806, indicating good internal consistency reliability for assessing GAD symptoms in this population [36].

##### World Health Organisation—Five Well-Being Index (WHO-5)

The WHO-5 is a concise self-reported tool used to evaluate current mental well-being [37]. It comprises five positively phrased items rated on a 6-point Likert scale, ranging from 0 (not at all) to 5 (all the time). The total raw score ranges from 0 to 25 and is multiplied by 4 to obtain the final score, where 0 represents the worst possible well-being, and 100 represents the best. Scores of 50 or less indicate poor well-being and warrant further assessment for possible depression symptoms, while scores of 28 or lower indicate depression. The measure has been found to have good construct validity as a unidimensional scale assessing well-being in both younger and older populations [37]. The Chinese version of the WHO-5 has been found to have good psychometric properties. The scale can serve as a valuable measure in epistemological studies and clinical research concerning well-being in Chinese populations. The WHO-5 is unidimensional and exhibits good internal consistency, with Cronbach’s alpha values of 0.85 and 0.81 [38].

### 3.5. Statistical Methods

#### Statistical Methods for Primary and Secondary Outcomes

The statistical analysis for this study will be conducted using IBM SPSS Statistics 26.0. Only participants who attend the entire 2 h group session will have their responses included in the analysis. Ineligible participants will be excluded, and the remaining ones will be analysed according to their initial randomisation. The characteristics of participants in both the intervention and waitlist control groups will be described in terms of frequency and percentage.

All analyses will be conducted according to the intention-to-treat principle [39]. Descriptive statistics will be used to ascertain any significant differences in demographic or clinical variables at baseline.

In all analyses, we will present regression coefficients for continuous outcomes, with 95% confidence intervals and *p* values. Linear regressions will be used to analyse the primary outcome, measured by the Inclusion of Other in the Self (IOS) Scale. These regressions will compare the intervention and control groups as randomised, adjusting for baseline IOS scores and any stratification or minimization variables [40]. The same approach will be applied to continuous secondary outcomes, comparing the groups at the post-intervention and one-month follow-up. All analyses will include adjustments for baseline measurements and relevant covariates to ensure accurate and robust comparisons.

Repeated-measures ANOVAs will be used to examine changes in outcomes over three time points. An interaction between the treatment group and time will determine whether treatment effects are presented and/or sustained. We will conduct sensitivity analyses to examine the impact of missing data, baseline imbalances in important prognostic factors, number of treatment sessions attended, potential “facilitator effects”, and the timing of questionnaire completion [41]. No planned interim analysis of outcome data will be conducted.

### 3.6. Participant Timeline

As shown in Figure 1, control group participants will undergo data collection at the same intervals (t_0_, t_1_, t_2_) as the intervention group, without receiving the intervention during this period. Upon completing data collection at t_2_, control group participants will be offered the intervention to ensure they also receive support”.

### 3.7. Sample Size

To determine the appropriate sample size for this study, we referred to Whitehead’s recommendations [42], which suggest that pilot trials require per-group sample sizes of 75, 25, 15, and 10 for standardised effect sizes (Cohen’s d) that are extra small (≤0.1), small (0.2), medium (0.5), and large (0.8), respectively. Additionally, prior ACL studies conducted by Tsai [21] and others [43] have shown that sample sizes of 30–35 participants per group are sufficient to detect medium effects in similar interventions.

Based on previous research [21], we expect a small to medium effect size for this study (Cohen’s d = 0.366, which corresponds to f ≈ 0.183). To ensure this study is adequately powered to detect this effect in an older adult population, we performed a power analysis using G*Power (version 3.1.9.7). We employed a repeated-measures ANOVA with a within–between interaction (two groups measured at three time points), setting the significance level (α) at 0.05 and the desired statistical power (1 − β) at 0.80 to minimise the risks of Type I and Type II errors.

The power analysis indicated that a minimum total sample size of 50 participants (25 per group) would be required to detect a significant difference between the intervention and control groups, assuming a small to medium effect size. To account for potential attrition over the course of this study, we increased the sample size by 15–20%, resulting in a target recruitment of approximately 60 participants (30 per group). This adjustment will enhance the robustness of our findings and ensure sufficient statistical power to evaluate the effectiveness of the ACL intervention among older adults.

### 3.8. Recruitment

The recruitment process will commence with referrals from local community organisations, employing purposeful convenience sampling and snowball sampling to ensure a diverse participant pool. The project leaflet was distributed at the community organisation and sent to potential participants who met the eligibility criteria. An online application form was made available for interested individuals to complete based on their willingness to participate. A research assistant will assess the eligibility of the applicants. Additionally, an online information session will be conducted for potential participants to support and evaluate their proficiency in using online platforms. To further support older participants’ use of online platforms, a step-by-step guideline will be provided. Eligible individuals will be provided with detailed information about this study, and informed consent will be obtained from those willing to participate.

### 3.9. Allocation and Blinding

A randomisation process will be implemented to ensure the equal distribution of participants between the intervention and waitlist control groups. An independent researcher will generate computer-generated random codes for group assignment.

The computer-generated random codes will be sealed in opaque envelopes. The envelopes will be sequentially numbered, and the randomisation process will remain blinded for participants throughout to prevent bias.

Participants will be notified by the research assistant via email or text about their assignment, and their willingness to engage in both the intervention and data collection will be confirmed.

Double-blind randomisation is not possible, because participants and facilitators of the online sessions will know automatically whether a participant is receiving the intervention. Therefore, we will apply single blinding, in which the outcome assessors and statisticians that oversee the assessments and analyse the results are blind to the condition. Randomisation will be performed by a research assistant who is not involved in the assessments. Randomisation outcomes will only be accessible to this research assistant. Participants will be instructed not to share their randomisation outcome with the researcher overseeing their assessments, and facilitators of the intervention will be instructed not to discuss identifying details of intervention participants with the researcher. Unblinding is possible if needed in the case of (serious) adverse events, and can be performed by the primary investigator, who is not involved in assessments.

To ensure that professional support is not withheld, due to participation in this study, participants in the waitlist control group will be invited to complete the same intervention after the study period, once all data collection (pre, post, and follow-up) has been completed. During the study period, the control group will undergo regular assessments at the same time points as the intervention group. This ensures that all participants receive the intervention while maintaining the integrity of the control group for comparison during the study period. The control group participants will be contacted and provided with access to the online session immediately following the final data collection point.

### 3.10. Data Collection and Management

Baseline Assessment (*t*_0_). Participants will undergo a comprehensive baseline assessment involving the completion of demographic surveys covering age, gender, marital status, educational level, employment status, and self-rated health. Simultaneously, baseline measurements, including the Inclusion of Other in the Self scale, 6-item de Jong Gierveld loneliness scale, PHQ-9, GAD-2, and WHO-5, will be administered.

Immediate Post-Intervention Assessment (*t*_1_). Immediately following the ACL session, participants will complete post-intervention measurements, mirroring the baseline assessments.

One-Month Follow-up Assessment (*t*_2_). One month after the intervention, participants will undergo follow-up measurements, mirroring the baseline assessments. This follow-up aims to capture the sustainability of the intervention effects. All assessments will be conducted through online surveys, maintaining participant confidentiality.

Participants who complete the questionnaires at all three time points will receive monetary incentives in the form of gift coupons. Participants would be briefed that their participation in this study is entirely voluntary, and the distribution of monetary incentives will not be in any way related to their responses given in this study.

Data will be managed through a secure online database maintained by Saint Francis University. Participant information will be stored in a portable drive where it will be locked in the filing cabinet at the university to ensure confidentiality. All personal identifying information will be removed from the research data within 24 months of study completion.

Upon consent, personal phone numbers will be collected for potential future studies. If consent is not given, only the last four digits of their phone numbers will be collected to facilitate data matching in the pre-test–post-test design. The interventions will be conducted via Zoom, and video recordings of the sessions will be used for evaluation and validity checks.

## 4. Discussion

Conducting a randomised controlled trial (RCT) study for older adults targeting loneliness and other mental health outcomes requires careful ethical considerations [44]. This study should comply with ethical principles and guidelines for research involving human subjects. The participants must be informed about this study’s aims, procedures, and possible risks and benefits before agreeing to participate. The participants should be selected based on relevant criteria, such as age, gender, and health status. The selection process should be unbiased, and the participants should be recruited from different sites. The researchers should ensure that the intervention does not harm the participants and that they receive adequate support throughout this study. The participants should also be given the option to withdraw from this study at any time without any penalty. This study should be designed in such a way that it does not discriminate against any group or individual. Finally, the researchers should ensure that this study’s findings are presented accurately and honestly, and that the participants’ privacy and confidentiality are protected at all times. These ethical considerations are crucial to ensure this study is conducted in a fair, safe, and respectful manner.

This proposed research explores the potential efficacy of a one-session online ACL intervention for reducing loneliness and depression/anxiety symptoms and enhancing social connectedness and subjective well-being among older adults. We also aim to estimate the effect sizes of the ACL intervention. This study can reach theoretical, empirical, practical, and teaching significance. This study enriches the theoretical understanding of Socioemotional Selectivity Theory by integrating the online component. Concerning the empirical aspect, this proposed research can empirically support the efficacy of the Chinese version of ACL for seniors using an experimental design, contributing to the growing literature on online interventions for mental well-being [45]. Regarding practical significance, the research team can further facilitate the training of practitioners to implement online psychological interventions for older adults. From the teaching perspective, exploring a new way to support the social and emotional well-being of older adults could inform undergraduate courses like Developmental Psychology and Positive Psychology Interventions. Findings from this study can update teachers’ understanding of psychological intervention for older adults, specifically the feasibility of using an online ACL intervention.

One primary limitation revolves around the potential challenges associated with the randomisation process. Despite employing random assignment techniques to allocate participants to either the intervention or control groups, there exists a possibility of inadvertent biases in the randomisation process [46]. Such biases could compromise the integrity and validity of the study outcomes. For instance, if the randomisation process lacks true randomness, certain individuals may inadvertently be predisposed to one group over the other, thereby introducing confounding variables. The utilisation of purposive convenience/snowball sampling for participant recruitment may also introduce additional limitations. While this approach was chosen due to the nature of our target population and data collection sources, it may inherently introduce a selection bias from the initial stages of participant recruitment. This bias may impact the generalizability of our findings, as the sample may not be representative of the broader population of older adults. To address this limitation, we will collaborate closely with NGOs to ensure the recruitment of participants with diverse profiles, thereby enhancing the representativeness of our sample. Another critical limitation pertains to the issue of imperfect blinding inherent in the study design. Given the nature of the intervention, the complete blinding of both participants and researchers to group assignments may not be feasible. This lack of blinding could potentially influence study outcomes, as participants may modify their behaviour based on their awareness of group assignments. To address this limitation, participants will be single-blinded regarding the main outcome measures throughout the study duration, minimising the likelihood of biassed responses. Additionally, participants will undergo debriefing sessions upon completion of data collection, fostering transparency and mitigating potential biases introduced by imperfect blinding [47].

## 5. Conclusions

The proposed protocol offers a significant and novel framework for investigating the impact of the online ACL intervention on social connectedness and other psychological indicators among Chinese older adults. This pilot RCT study will have potential for meaningful contributions to the further investigation and development of online interventions for older adults. This study can reach theoretical, empirical, and practical significance.

## Figures and Tables

**Figure 1 healthcare-12-02158-f001:**
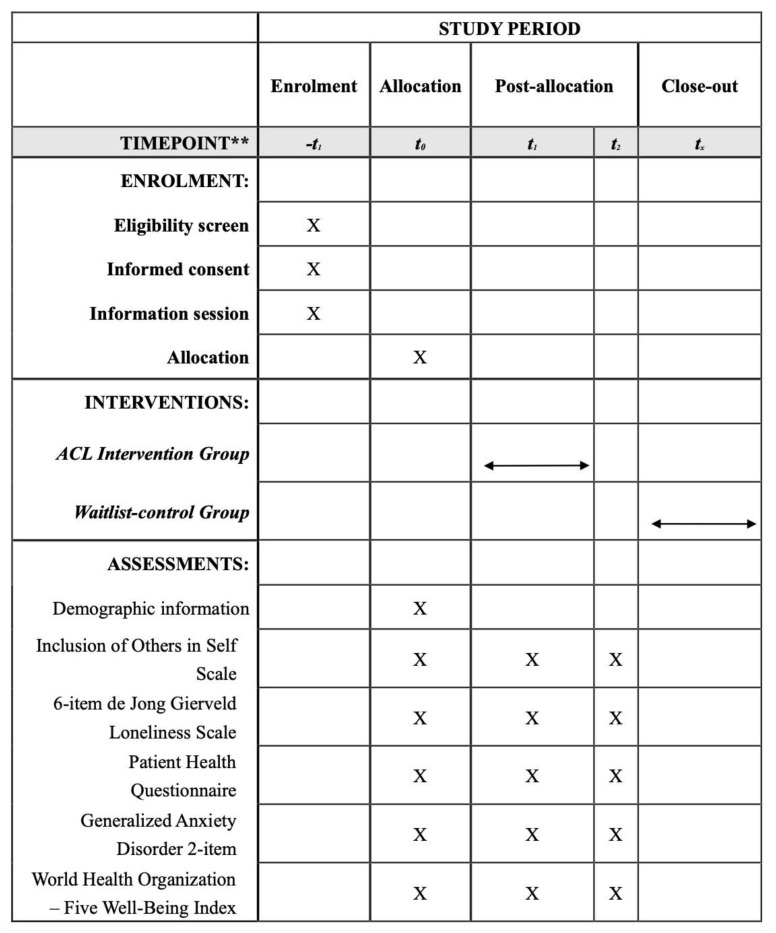
The schedule of enrolment, interventions, and assessments. *t*_0_ = baseline assessment; *t*_1_ = immediate post-intervention assessment; *t*_2_ = one-month follow-up assessment. “X” indicates specific actions or procedures occurring at the designated time points, “
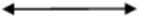
” indicates the duration of the interventions.

**Table 1 healthcare-12-02158-t001:** Intervention protocol.

Protocol Section	Content	Time
Introduction and Welcome	Participants and facilitators introduce themselves; an overview of ACL principles.	12 min
Shared Agreements	Establish and explain group rules using the “five-finger agreement” method.	5 min
Theme Introduction and Mindful Drinking	Introduction of the meeting theme “making friends and being seen”. Participants engage in a mindful drinking exercise.	15 min
Meditation	Guide a five-finger breathing exercise; reflect on life stories and experiences.	8 min
Meditation Debriefing	Participants share their present-moment feelings using emotional metaphors (sweet, sour, bitter, spicy).	5 min
Contemplation Questions	Participants reflect on one of three contemplation questions and are invited to share their thoughts on their life experiences.	5 min
Modelling the Sharing Process	Facilitators model the sharing process before small group sharing.	5 min
Sharing Process	Participants are divided into sub-groups and take turns sharing and receiving feedback in the breakout rooms.	22 min
Debriefing	Sharing about the breakout room experience; use emotional metaphors if needed.	12 min
Wrap Up and Feedback	Participants share their experiences of the session, followed by reminders about the follow-up assessment.	6 min
Post-Intervention Assessment	Completion of post-intervention questionnaires online.	20 min

## Data Availability

The datasets presented in this article are not readily available, because the data are part of an ongoing study.

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
