# Peer review of "An Awareness, Courage, and Love Online Group Intervention for Chinese Older Adults in the Post-Pandemic Era: Study Protocol for a Randomised Controlled Trial"

_healthcare, 2024, doi:10.3390/healthcare12212158_

Round 1

Reviewer 1 Report

Comments and Suggestions for Authors

My main concern stays the same: the manuscript is a plan. It is difficult to evaluate a study when the only thing we have is the planning.     

Plus, the authors adopt the Socioemotional Selectivity Theory (SST) as the theoretical framework to guide their study planning. The theory contends that, as people age, they would experience age-related losses (changes in social networks), but older adults would try to maintain their well-being by prioritizing their close relationships over peripheral ones. At the same time, the Convoy Model of Social Relations argues that changes in social relationships is a life course process. It takes time for them to start adjusting their core "convoys" along the way. So it is important to maintain a longitudinal perspective. Is this study taking the changes in social network" into account? How old should a participant be recruited to the study? Should a participant be 60 or 70 years old in order to experience some significant social network changes? People are different and older people are especially so.

Reviewer 2 Report

Comments and Suggestions for Authors

The study adds significant importance to the field and has a well-designed protocol addressing an important topic. Addressing these points could underscore the importance of the study and its potential impact.​​​​​​​​​​​​​​​​

Methods:

- The single-session format may be a limitation. Consider discussing the rationale for this approach versus a multi-session intervention.

- The computation of the sample size might be more thorough. Give the power, alpha level, and predicted effect size that were used in the G*Power analysis.

- Clarify how the waitlist control group will be handled - will they receive the intervention after the study period?

- Indicate how missing data will be handled (multiple imputation, for example).

- Take into account including subgroup analysis based on pertinent demographic variables.

Ethical Considerations:

- Address how you will handle participants who show signs of severe depression or anxiety during screening.

Overall 

- Consider adding a flowchart to illustrate the study design and participant flow.

Reviewer 3 Report

Comments and Suggestions for Authors

This protocol proposed to investigate the impact of ACL (Adult Cognitive Learning) on social loneliness and mental health in older adults, offering important insights into the role of psychiatric interventions in enhancing psychological well-being in aging populations.

There are areas for improvement that could strengthen the study's rigor. 

First, the authors should provide a clear explanation of how the sample size of 70 participants was determined. Specifically, they should consider using statistical methods such as an effect size estimation to justify the adequacy of the sample size for detecting meaningful differences or effects in the study.

Additionally, the manuscript would benefit from a detailed report on the reliability and validity of the scales used to measure the study's key outcomes. Demonstrating that the tools used to assess mental health and social loneliness are both reliable and valid is crucial for the study.

Finally and most importantly, the research design, which includes a one-time, two-hour intervention, should be better supported by existing literature. It is important to provide evidence that such a brief intervention can produce significant and lasting psychological and mental health benefits in older adults. Including more references to previous studies that have successfully implemented similar short-term interventions will help substantiate the study's approach.

Reviewer 4 Report

Comments and Suggestions for Authors

*Introduction:

The introduction lacks a clear explication of the direct connection between Socioemotional Selectivity Theory (SST) and the Awareness, Courage, and Love (ACL) model. While both models are described, their interrelation and the suitability of the ACL approach for older adults within the SST context require more explicit logical linkage. The following points could strengthen this connection:

A. SST posits that older adults focus more on emotionally meaningful experiences and relationships; the ACL model provides specific methods for forming and strengthening such meaningful relationships.

B. The 'Awareness' component of the ACL model can be associated with the enhanced emotional regulation abilities of older adults emphasized in SST.

C. The 'Courage' element of ACL relates to the self-disclosure necessary for forming deeper, more meaningful relationships within the reduced social networks of older adults, as mentioned in SST.

D. The 'Love' aspect of ACL directly correlates with the pursuit of emotional satisfaction and well-being in older adults, as proposed by SST.

The literature review on the application of the ACL model to older adult populations is limited. A more comprehensive examination and presentation of existing research on the effectiveness, applicability, and potential benefits of ACL interventions for older adults is necessary.

The cited research on online applications of the ACL model is limited to studies with general adult couples. This paucity of relevant research could be a limitation. It is recommended to include discussions on the effectiveness of online psychological interventions for older adults in general, or the efficacy of similar online intervention programs.

*Study Aims and Hypotheses:

It is advisable to include aims or hypotheses that consider the specific characteristics of online interventions. For instance, hypotheses regarding older adults' competency or satisfaction with online platforms could be incorporated.

While a hypothesis on the maintenance of effects after one month is present, including aims or hypotheses about more long-term effects could enhance the clinical significance of the study.

Adding hypotheses directly related to SST or the ACL model could strengthen the theoretical foundation of the research.

*Materials and Methods:

A more detailed explanation of the sample size calculation rationale and methodology would be beneficial. Including expected effect sizes and power calculations would strengthen this section.

The recruitment methods and strategies for older participants should be more explicitly described, particularly clarifying how older adults capable of participating in online interventions will be identified.

A more detailed explanation of the specific content and process of the ACL sessions would be valuable.

Describe plans for supporting older participants' use of online platforms (e.g., technical support, pre-training).

Additional mention of specific ethical considerations related to older adults' online participation (e.g., digital literacy, online fatigue) would be prudent.

Clear delineation of the specific treatment content and schedule for the control group is necessary.

*Discussion:

While the main limitations such as potential bias in the randomization process, representativeness issues in sample selection, and imperfect blinding are mentioned, specific concerns about older adults' participation in online interventions are not addressed. Despite the inclusion criterion of "access to the internet and a videoconferencing device," concerns or limitations regarding the application of online interventions to older adults need to be resolved. This could be addressed by considering additional participant selection criteria, such as assessing older adults' digital literacy levels. Alternatively, explicit inclusion of pre-training or support plans for online platform use by study participants should be considered.

Reviewer 5 Report

Comments and Suggestions for Authors

Title: There is a slight gap in the sentence

Line 49-52: A little more context needed, explanation why older adults levels of loneliness, anxiety and depression increased following relaxation of restriction policy. The pandemic restrictions have not been in place for some time, why the urgency in expanding MHC interventions= more context.

Table 1 is well constructed and clear for the reader. Methodology, sampling and strategy is concise and appropriate for the study. This study protocol has a very clear focus and strong evidence base surrounding the development of the ACL model and no doubt the findings should be duplicated in other international studies. Although the pandemic and restrictions has passed, are their other considerations or cultural factors that you may consider for older adults loneliness, perhaps family connections, role of children in meeting older adults needs; just for your consideration. 

Reviewer 6 Report

Comments and Suggestions for Authors

Thank you for the opportunity to review.

I read it with great interest.

I have a few concerns, please see below.

About the title

The intervention method, "An Awareness, Courage, and Love Online Group Intervention," seemed abstract and vague. Since the subtitle says "Randomized Controlled Trial," I assume that this is an evidence-based research design. This makes me more concerned about the abstract nature of the main title.

In addition, I did not understand "An Awareness, Courage, and Love Online Group Intervention" just by reading the abstract.

Please revisit this point as well.

About the target audience

I believe that subjects who can participate in online interventions are better able to maintain social connections than those who cannot. I believe that subjects who are unable to use the Internet and have low information literacy are at greater risk. This point is important when designing a research protocol. The authors' thoughts should be clearly stated.

About Appendix A

The "Adapted Inclusion of Other in the Self (IOS) Scale" is written in Chinese.

I would like to see an additional explanation in English.

That is all.

Round 2

Reviewer 1 Report

Comments and Suggestions for Authors

This is a well-written proposal. I don't have further comments.

Author Response

Thanks for the reviewer's efforts.

Reviewer 6 Report

Comments and Suggestions for Authors

I believe that appropriate corrections have been made to the manuscript.

That is all.

Author Response

Thanks for the reviewer's efforts.